Toxic effects of sodium dodecyl sulfate on planarian Dugesia japonica

Feng Minmin
Xu Zhenbiao
Yin Dandan
Zhao Zelong
Zhou Xiuyuan
Song Linxia slxch@163.com
School of Life Sciences and Medicine, Shandong University of Technology , Zibo , China
Gao Junkuo
Electronic publication date: 2023 Jul 10
Publication date: 2023
Volume: 11
Electronic Location ID: e15660
Received 2023 Feb 14; Accepted 2023 Jun 7
Copyright: ©2023 Feng et al.
Copyright year: 2023
Copyright holder: Feng et al.
License: This is an open access article distributed under the terms of the Creative Commons Attribution License, which permits unrestricted use, distribution, reproduction and adaptation in any medium and for any purpose provided that it is properly attributed. For attribution, the original author(s), title, publication source (PeerJ) and either DOI or URL of the article must be cited.
License URL: https://creativecommons.org/licenses/by/4.0/

Keywords: Sodium dodecyl sulfate, Planarian, Oxidative stress, RAPD, qPCR

Funding: National Natural Science Foundation of China 31550005 31350004 Natural Science Foundation of Shandong Province, China ZR2011HM065 This work was supported by the National Natural Science Foundation of China (31550005, 31350004) and the Natural Science Foundation of Shandong Province, China (ZR2011HM065). The funders had no role in study design, data collection and analysis, decision to publish, or preparation of the manuscript.

==============================
Sodium dodecyl sulfate (SDS) is an anionic surfactant, which is widely used in various fields in human life. However, SDS discharged into the water environment has a certain impact on aquatic organisms. In this study, planarian Dugesia japonica (D. japonica) was used to identify the toxic effects of SDS. A series of SDS solutions with different concentrations were used to treat planarians for the acute toxicity test , and the results showed that the semi-lethal concentration (LC50) of SDS to D. japonica at 24 h, 48 h, 72 h, and 96 h were 4.29 mg/L, 3.76 mg/L, 3.45 mg/L, and 3.20 mg/L respectively. After the planarians were exposed to 0.5 mg/L and 1.0 mg/L SDS solutions for 1, 3, and 5 days, the activities of superoxide dismutase (SOD), catalase (CAT), and malondialdehyde (MDA) content were measured to detect the oxidative stress and lipid peroxidation in planarians. Random amplified polymorphic DNA (RAPD) analysis was performed to detect the genotoxicity caused by SDS to planarians. The results showed that the activities of SOD, CAT, and MDA content increased after the treatment, indicating that SDS induced oxidative stress in planarians. RAPD analysis showed that the genomic template stability (GTS) values of planarians treated by 0.5 mg/L and 1.0 mg/L SDS for 1, 3, and 5 days were 67.86%, 64.29%, 58.93%, and 64.29%, 60.71%, 48.21%, respectively. GTS values decreased with the increasing of SDS concentration and exposure time, indicating that SDS had genotoxicity to planarians in a time and dose-related manner. Fluorescent quantitative PCR (qPCR) was used to investigate the effects of SDS on gene expression of planarians. After the planarians were exposed to 1.0 mg/L SDS solution for 1, 3, and 5 days, the expression of caspase3 was upregulated, and that of piwiA, piwiB, PCNA, cyclinB, and RAD51 were downregulated. These results suggested that SDS might induce apoptosis, affect cell proliferation, differentiation, and DNA repair ability of planarian cells and cause toxic effects on planarian D. japonica.

Introduction

Surfactants are a kind of amphipathic compounds, which are widely used in our everyday life. Their global use is increasing every year, with consumption expected to reach $28.8 billion by 2023 (Brycki et al., 2017; Kaczerewska et al., 2020). Sodium dodecyl sulfate (SDS) is a common anionic surfactant with the chemical formula CH3(CH2)11OSO3Na, which is easily soluble in water and has the ability of decontamination, emulsification, and foaming (Cao et al., 2020). SDS is widely used in the production of cosmetics and cleaning products such as soaps, shampoos, shower gels and toothpastes (Bondi et al., 2015; Cao et al., 2020). The content of SDS varies depending on the product type and manufacturer, typically ranges from 0.01% to 50% in cosmetic products and 1% to 30% in cleaning products (Bondi et al., 2015). Consumers expose to SDS through the use of products containing this ingredient, and abuse of products may cause skin inhalation contact and skin inflammation (Bondi et al., 2015; Cao et al., 2020). SDS is also used in pharmaceutical and food products, it can be used as a food or drug additive to solubilize hydrophobic aromas or some types of preservatives (Cid et al., 2019). In addition, as a tissue lysate and protein denaturant, SDS can form complex with protein through hydrophobic interaction, so it is often used in Western blot, Chromatin Immunoprecipitation, SDS-PAGE, and other experiments in the field of biological research (Al-Tubuly, 2000; Brunelle & Green, 2014; Lai et al., 2017).

Although most surfactants are degradable, their continuous use and excessive emissions have caused pollution to the water environment (Bhattacharya et al., 2022; Lechuga et al., 2016; Mustapha & Bawa-Allah, 2020; Rosety-Rodríguez et al., 2002). The discharge of cleaning products containing SDS into the water environment through domestic wastewater had toxic effects on aquatic organisms (Bondi et al., 2015; Cruzde Carvalho et al., 2022; Jönander, Backhaus & Dahllöf, 2022; Messina et al., 2014). SDS in the water entered the fish body through gills, skin, or intestinal epithelial cells, circulated to various parts of the body, interrupted the normal steroidogenesis process, and reduced the production of sex hormone (Moniruzzaman & Saha, 2021; Rosety-Rodríguez et al., 2002). SDS was toxic to fish and sea urchin; the 96 h LC50 of SDS to Tigriopus fulvus, Dicentrarchus labrax, Dunaliella tertiolecta, and Paracentrotus lividus were 7.42 mg/L, 7.34 mg/L, 4.80 mg/L, and 3.20 mg/L, respectively (Bondi et al., 2015; Mariani et al., 2006).

The antioxidant defense system can protect organisms from oxidative damage caused by external pollutants. Superoxide dismutase (SOD) and catalase (CAT) are important components of the antioxidant defense system of organisms, and their activities can reflect the antioxidant level of organisms (Liu et al., 2021; Zhang et al., 2022). Malondialdehyde (MDA) is one of the degradation products of lipid peroxidation, it can be used as a biomarker of oxidative stress to detect the degree of oxidative stress of organisms (Tsikas, 2017). SDS could significantly increase the activities of SOD, CAT, and MDA content in Cirrhinus cirrhosus, Heteropneustes fossilis, and Tubifex tubifex, causing lipid peroxidation and oxidative stress in the organisms (Bhattacharya et al., 2021; Moniruzzaman & Saha, 2021). The toxic effects of SDS might be related to the alteration of cellular ionic balance caused by the changes of cellular membrane permeability and the induction of oxidative stress (Messina et al., 2014).

Planarian is a representative animal of the phylum Platyhelminthes, it is widely distributed in clean waters around the world (Zhang et al., 2016a). Due to its high chemical sensitivity, many chemical pollutants could cause toxic effects on planarians, resulting in the changes of locomotion, regeneration, neurotransmission, and even chromosome (Lau et al., 2007; Ofoegbu et al., 2016; Prá et al., 2005; Rink, 2013; Simão et al., 2020; Yuan et al., 2018; Zhang et al., 2016a). Planarian has become one of the indicator organisms for assessing the toxicity of environmental pollutants in the field of neuropharmacology and ecotoxicology (Buttarelli, Pellicano & Pontieri, 2008; Hagstrom et al., 2015; Prá et al., 2005; Wu & Li, 2018). Therefore, we used planarian as the test animal to study the toxic effects of SDS on aquatic organisms.

Randomly amplified polymorphic DNA (RAPD) is a technique for polymorphism analysis of genomic DNA, which is often used for the detection of genetic diversity and genotoxicity analysis (Pandey, Nagpure & Trivedi, 2018; Zare et al., 2019; Zhou et al., 2011). It is an important method for detecting the genotoxicity of drugs and pollutants to planarians (Yin et al., 2022; Zhang et al., 2016b). However, the application of RAPD technology for detecting the genotoxicity of SDS to planarians has not been reported. In this study, we analyzed the acute toxicity, oxidative stress reaction and genotoxicity of SDS to planarians. The effects of SDS on the expression of genes related to cell apoptosis, proliferation, differentiation, and DNA repair ability were further detected by fluorescent quantitative PCR (qPCR). Our results will provide theoretical basis for the research of the ecotoxicity of SDS to aquatic organisms, and provide theoretical reference for the management and protection of freshwater ecosystems.

Materials & Methods

Materials

SDS was purchased from Biosharp Company of China, and the purity was 99%. Total protein quantitation kit, SOD, CAT, and MDA test kits were purchased from Nanjing Jiancheng Company of China for determination of the activities of SOD, CAT and MDA content. The E.Z.N.A.® Mollusc DNA kit was the product of Omega Bio-Tek Company for extraction of genomic DNA, and 2 × Taq PCR StarMix was purchased from GenStar Company. Trizol reagent was purchased from Thermo Fisher Technology Co., Ltd. for extraction of RNA. Reverse Transcription kit and TB Green premix Ex Taq II (2×) were purchased from TaKaRa Company. The sequences of 13 random primers and qPCR primers used in this study are shown in Tables S1 and S2.

Test animals

Planarians used in this experiment were the asexual strain Dugesia ZB-1, which were cultured in the laboratory in Montjuïc water (1.6 mmol/L NaCl, 1.0 mmol/L CaCl2, 1.0 mmol/L MgSO4, 0.1 mmol/L MgCl2, 0.1 mmol/L KCl, and 1.2 mmol/L NaHCO3) in a biochemical incubator (SPX-2508SH, Shanghai CIMO Medical Instrument Manufacturing Co., Ltd., China) at 20 °C. Animals were fed with beef liver twice a week and starved for a week before the experiment.

Acute toxicity test

Based on reference (Li, 2008) and pre-experimental results, planarians with body length of about one cm were exposed to SDS solutions of six different concentrations (3.0 mg/L, 3.5 mg/L, 4.0 mg/L, 4.5 mg/L, 5.0 mg/L, and 6.0 mg/L), and the control group were cultured in Montjuïc water. 10 planarians in each group were treated in Petri dishes with a volume of 10 mL solution. SDS solutions were renewed and the mortality of planarians were calculated at an interval of 24 h. The experiment was repeated three times to prevent accidental error. In order to obtain the relationship between the concentration of SDS and the mortality of planarians, regression equation was obtained by plotting the logarithm of concentration and odds unit. Each time point corresponded to a regression equation. Odds unit with mortality rate of 50% was taken into the regression equation and the LC50 of 24 h, 48 h, 72 h, and 96 h were calculated (Hagstrom et al., 2015).

Detection of antioxidant enzyme activity

In order to reflect the oxidative stress response and the oxidative damage to planarians under SDS stress, the activities of antioxidant enzymes and MDA content were detected (Gao et al., 2022; Wang et al., 2020). 10 planarians exposed to 0.5 mg/L and 1.0 mg/L SDS solutions for 1, 3, and 5 days were used to detect antioxidant enzyme activities. The planarians cultured in Montjuïc water were as control. After the exposure, 10 planarians were put in a precooled mortar and two mL PBS buffer was added to grind them to paste. The paste was centrifugated at 4 °C, 10,000 rpm for 10 min and the supernatant was used to detect the protein concentration, the activities of SOD, CAT, and MDA content according to the instructions of the corresponding test kits. After adding the corresponding reaction reagents provided by the kits, the absorbance values were measured at the wavelength of 550 nm, 405 nm, and 532 nm, respectively. Finally, the activities of SOD, CAT, and MDA content were calculated based on the absorbance values.

Genomic DNA extraction

Genomic DNAs of planarians exposed to SDS at concentrations of 0.5 mg/L and 1.0 mg/L for 1, 3, and 5 days were extracted according to the instructions of DNA kit. The integrity of DNA was detected by 1% agarose gel electrophoresis, and the purity and concentration were measured by detecting OD260/OD280 with a micro-spectrophotometer (K5600; Beijing Kaiao Technology Development Co., Ltd., Beijing, China).

RAPD amplification

RAPD can be applied to detect the changes in genomic DNA at the molecular level (Zhang et al., 2016b), so the method was used for the genotoxicity assay in this study. Each polymerase chain reaction (PCR) was conducted in a mixture of 25 µL containing 20 ng genomic DNA, 0.2 µmol/L primer, and 12.5 µL 2 × Taq PCR StarMix. Amplifications were carried out in a DNA thermocycler (TC-XP, Hangzhou Bioer Technology Co., Ltd., China).The PCR program was 94 °C for 5 min, 40 consecutive cycles including 94 °C for 1 min, 37 °C for 1 min, and 72 °C for 2 min, then followed by 72 °C for 10 min as the final extension (Zhang et al., 2016b). After amplification, the PCR products were analyzed by electrophoresis on 1% agarose gel at a voltage of 100 V and a current of 200 mA for 60 min. Then the electropherograms were photographed under an AlphaImager HP system (Alpha2200-5; Alpha Innotech, San Leandro, CA, USA).

Estimate of GTS

Genomic template stability (GTS) is an indicator of genotoxicity, and its value can reflect the degree of genotoxicity. GTS is calculated according to the formula: GTS%=1−an×100, where “a” represents the number of polymorphic bands detected in each treatment sample; “n” represents the number of total bands in the control (Atienzar et al., 1999; Zhang et al., 2016b). The GTS of the control group is set to 100%, and the GTS of each treatment group is expressed as a percentage of the control group.

RNA extraction and qPCR

After exposure to 1.0 mg/L SDS for 1, 3, and 5 days, the RNAs of planarians of the treatment groups and the control group were extracted. Reverse transcription and qPCR were conducted to study the expression level of genes (Liang et al., 2022). Total RNA was extracted by Trizol reagent and was reverse transcribed into cDNA with the reverse transcription mixtures of 20 µL containing 2 µg RNA and 2.5 µmol/L oligo(dT)15 as primer. The qPCR was performed in a Light Cycler 480 System (Roche Diagnostics, Basel, Switzerland) with the PCR program was 95 °C for 30 s followed by 40 consecutive cycles consisting of 95 °C for 5 s, 58 °C for 10 s, and 72 °C for 15 s. Each group was performed in triplicate. The relative expression levels were calculated using the 2−ΔΔCt method with Dj-Actin gene as the endogenous standardization.

Statistical analysis

Statistical analysis and regression analysis were performed using SPSS 26.0 software. The odds unit with the mortality rate of 50% was taken into the regression equation to calculate the LC50 and 95% confidence interval (95% CI) of SDS to D. japonica. R2 represents the coefficient of determination. The activities of SOD and CAT, MDA content, and the levels of gene expression were presented as mean ± SD. One-way ANOVA was used to compare the differences between each treatment group and the control group. The value of p < 0.05 was considered statistically significant, and p < 0.01 represents highly significant (Gao et al., 2022; Liang et al., 2022).

Results

Acute toxicity of SDS to planarians

After acute toxicity experiment, the regression equation, R2, LC50 and 95% confidence interval (95% CI) of SDS to D. japonica were calculated. As shown in Table 1, the LC50 of SDS to D. japonica at 24 h, 48 h, 72 h, and 96 h were 4.29 mg/L, 3.76 mg/L, 3.45 mg/L, and 3.20 mg/L respectively. Results showed that the acute toxicity of SDS to planarians increased with the extension of exposure time.

Table 1 Semi-lethal concentration (LC50) of SDS to D. japonica at different time points.

Exposure time (h)	Regression equation	R2	LC50 (mg/L)	95% CI (mg/L)	
24	y = 11.6x − 2.36	0.961	4.30	4.09∼4.51	
48	y = 13.5x − 2.76	0.941	3.76	3.61∼3.93	
72	y = 11.5x − 1.19	0.998	3.45	3.28∼3.63	
96	y = 13.3x − 1.71	0.955	3.20	3.06∼3.34	
Notes.

x is the logarithm of concentration, y is the odd unit; R2 is the “coefficient of determination”; 95% CI is 95% confidence interval.

Effects of SDS on oxidative stress

After treatment with 0.5 mg/L and 1.0 mg/L SDS solutions for 1, 3, and 5 days, the activities of SOD, CAT, and MDA content in planarians were detected. Results showed that the activities of SOD, CAT, and MDA content in the treatment groups were higher than that of the control group. With the extension of SDS exposure time, SOD activity in 0.5 mg/L SDS treatment groups increased significantly, reached the highest on the fifth day. In the 1.0 mg/L SDS treatment groups, SOD activity highly significantly increased on the first day, continued to increase on the third day, and returned to the same level as the control group on the fifth day (Fig. 1A). The CAT activities of the 0.5 mg/L and 1.0 mg/L SDS treatment groups significantly increased firstly from the first to the third day, and then decreased from the third to the fifth day (Fig. 1B). The MDA content in the 0.5 mg/L SDS treatment groups showed an increasing trend from the first to the fifth day, while in the 1.0 mg/L SDS treatment groups, it was highest on the first day, and then decreased, reaching the lowest level on the fifth day (Fig. 1C). These results suggested that the treatment of SDS might cause oxidative stress and lipid peroxidation in planarians.

Figure 1 The effects of SDS on oxidative stress of D. japonica.

(A) SOD, (B) CAT activities and (C) MDA content of the planarians exposed to 0.5 mg/L and 1.0 mg/L SDS for 1, 3, and 5 days. *p < 0.05; **p < 0.01.

Effects of SDS on RAPD profiles

The OD260/OD280 value of each group of the planarians genomic DNA was between 1.7 and 2.2, and a single band was obtained by 1% agarose gel electrophoresis (Fig. 2A), indicating that the purity and integrity of DNA was good and no degradation. The bands of PCR products amplified with the same template and the same primer were consistent (Fig. 2B), indicating that this technique is repeatable.

Figure 2 Genomic DNAs and RAPD profiles of planarian D. japonica.

(A) Genomic DNAs isolated from D. japonica exposed to 0.5 mg/L and 1.0 mg/L SDS for 1, 3, and 5 days. M is 1 kb DNA ladder (10,000, 8,000, 6,000, 5,000, 4,000, 3,000, 2,000, 1,500, 1,000, 500 bp from top to bottom). 0 is control. (B) Reproducibility of RAPD profiles generated from D. japonica of the control group DNAs. M is DL2000 DNA marker (2,000, 1,000, 750, 500, 250, 100 bp from top to bottom). (C–F) RAPD profiles of genomic DNAs from D. japonica exposed to SDS using primers S5, S8, S10, and S17. M is DL2000 DNA marker.

The amplified products of RAPD were subjected to agarose gel electrophoresis, and obvious bands were obtained. A total of 56 bands were amplified from the genomic DNAs of the control group with 13 random primers, and 1∼9 bands were amplified with each primer (Table 2, Figs. 2C–2F). The number of polymorphic bands in 0.5 mg/L and 1.0 mg/L SDS treatment groups were 18, 20, 23, and 20, 22, 29 after 1, 3, and 5 days of exposure (Table 2), indicating that the RAPD patterns of SDS treated groups were different from that of the control group, and changed with the SDS concentrations and the exposure time.

Table 2 Analysis and statistics of different bands in RAPD profiles of control group and treatment groups.

Primer	Control	0.5 mg/L	1.0 mg/L	
		1d	3d	5d	1d	3d	5d	
		a	b	c	d	a	b	c	d	a	b	c	d	a	b	c	d	a	b	c	d	a	b	c	d	
S5	3	3	0	2	1	3	0	2	1	2	1	2	0	3	0	2	1	3	0	2	1	2	1	2	0	
S8	2	3	0	1	0	3	0	1	0	2	0	1	1	3	0	1	0	3	0	1	0	3	0	1	0	
S10	5	2	0	2	1	2	0	2	2	2	0	3	1	2	0	3	0	2	0	3	0	0	1	2	2	
S15	2	0	0	0	0	1	0	0	0	0	0	1	0	1	0	0	0	0	0	1	0	0	1	0	0	
S17	5	2	0	2	0	2	0	1	1	3	0	1	1	0	0	1	1	0	0	1	1	3	1	3	1	
S18	5	0	1	2	0	0	1	2	0	0	1	2	0	0	2	1	0	0	1	3	0	0	1	2	1	
S20	5	0	3	0	1	0	1	2	1	0	2	1	1	1	1	1	0	0	4	0	0	0	4	0	0	
S64	4	0	0	1	0	0	0	0	0	0	1	1	1	0	0	1	0	0	0	0	0	0	3	0	1	
S75	6	0	1	0	0	0	1	1	0	0	1	0	0	1	1	0	0	0	1	0	2	0	2	0	2	
S78	4	0	0	1	0	0	0	3	0	0	1	2	0	0	0	4	0	0	0	2	0	0	2	0	2	
S80	5	1	0	3	0	1	0	2	0	0	3	1	2	2	0	2	0	3	0	0	2	0	0	2	1	
S83	4	0	0	3	0	1	1	2	0	1	0	0	3	0	1	2	0	1	1	1	0	0	1	0	3	
S84	6	0	2	2	0	0	3	1	0	0	3	2	0	0	2	3	0	0	3	3	0	0	4	0	2	
Total	56	11	7	19	3	13	7	19	5	10	13	17	10	13	7	21	2	12	10	17	6	8	21	12	15	
a + b		18			20			23			20			22			29			
a + b + c + d		40	44	50	43	45	56	
Notes.

(a) Appearance of new bands; (b) disappearance of normal bands; (c) increase in band intensities; (d) decrease in band intensities; (a + b) polymorphic bands; (a + b + c + d) varied bands.

GTS is the percentage of the number of polymorphic bands in the RAPD maps to the total number of bands in the control group. The polymorphic bands amplified with each primer varied with concentration of SDS (0.5 mg/L, 1.0 mg/L) and exposure time (1, 3, and 5 days). The RAPD maps were analyzed and the GTS values were calculated. Results showed that the GTS values of planarians exposed to 0.5 mg/L and 1.0 mg/L SDS for 1, 3, and 5 days were 67.86%, 64.29%, 58.93%, and 64.29%, 60.71%, 48.21%, respectively (Fig. 3). GTS decreased with the increase of SDS concentration and the extension of exposure time, indicating that SDS has genotoxicity to planarians in a dose and time-related manner.

Figure 3 Genomic template stability (GTS) of D. japonica exposed to 0.5 mg/L and 1.0 mg/L SDS for 1, 3, and 5 days.

Effects of SDS on gene expression

The changes of gene expression in planarians were detected by qPCR after treatment with 1.0 mg/L SDS solution for 1, 3, and 5 days. Results showed that the expression level of apoptosis marker gene Dj-caspase3 was highly significantly upregulated, reaching the highest on the fifth day (Fig. 4A). The expression levels of cell proliferation related gene Dj-piwiA and neoblast differentiation related gene Dj-piwiB had no significant difference with the control on the first day, but decreased significantly on the third and the fifth days (Figs. 4B–4C). The cell proliferation marker gene Dj-PCNA and cell cycle related gene Dj-cyclinB significantly decreased from the first day, and reached the lowest on the fifth day (Figs. 4D–4E). The expression level of DNA damage related gene Dj-RAD51 significantly downregulated, and reached the lowest on the third day (Fig. 4F). The qPCR results showed that SDS might induce apoptosis, affect cell proliferation, differentiation, normal progression of cell cycle, and DNA repair ability in planarians.

Figure 4 Expression levels of mRNA by qPCR after exposure to 1.0 mg/L SDS for 1, 3, and 5 days in D.japonica.

The expression levels of (A) apoptosis related genes Dj-caspase3; (B) cell proliferation related gene Dj-piwiA, (C) neoblast differentiation related gene Dj-piwiB; (D) cell proliferation marker gene Dj-PCNA; (E) cell cycle related gene Dj-cyclinB; (F) DNA damage related gene Dj-RAD51. *p < 0.05; **p < 0.01.

Discussion

In this study, LC50 of SDS to planarian D. japonica was determined by acute toxicity test. Previous study showed that LC50 of SDS to planarian D. japonica was 0.36 mg/L, the same value from 24 h to 96 h (Li, 2008). In order to accurately measure the LC50 of SDS to D. japonica at different times, we conducted this acute toxicity experiment and obtained the corresponding values at different time points. Our results showed that LC50 was 3.20 ∼4.29 mg/L from 24 h to 96 h, and it was getting lower with the extension of SDS exposure time. Li (2008) determined LC50 of many surfactants to D. japonica, and the toxicity rank of 96 h LC50 was as follows: SDS > CTAB > NP > LAS > Hyamine 1622 > Triton X-100 > PFOS > PFOA. SDS had different toxicity to different aquatic organisms. Nunes, Carvalho & Guilhermino (2005) determined LC50 of SDS to three aquatic species including the euryhaline fish Gambusia holbrooki, the hypersaline crustacean Artemia parthenogenetica, and the marine algae Tetraselmis chuii. LC50 of SDS to A. parthenogenetica at 48 h was 12.2 mg/L, and that to T. chuii and G. Holbrooki at 96 h were 30.2 mg/L and 15.1 mg/L (Nunes, Carvalho & Guilhermino, 2005). In our study, LC50 of SDS to planarian D. japonica at 48 h and 96 h were 3.76 mg/L and 3.20 mg/L, indicating that the toxicity of SDS to planarian D. japonica was higher than that to the above three organisms.

When organisms are subjected to environmental stress, oxidative stress reactions usually occur and reactive oxygen species (ROS) is produced (Lushchak, 2016; Yang, Lim & Song, 2020). ROS is a substance with active properties and strong oxidizing power, its excessive accumulation can destroy the spatial structure of biological macromolecules such as DNA, proteins, and lipids, causing DNA damage or cell death (Finkel, 2011; Sachdev et al., 2021; Tang et al., 2019; Tsikas, 2017). The disequilibrium between the ROS formation and the neutralization by antioxidant enzymes can lead to the reactions of oxidative stress (Bhattacharya et al., 2022; Kurutas, 2016). SOD is the key enzyme that catalyzes the conversion of superoxide anion free radicals into hydrogen peroxide, while CAT converts hydrogen peroxide into water and molecular oxygen (Ding et al., 2009; Li et al., 2021). Therefore, SOD and CAT are the first line of defense for the organisms against oxygen toxicity, and the increase of their activities indicates that the antioxidant defense ability of organisms is enhanced (Wang et al., 2020; Zagal & Mazmanci, 2011). MDA is one of the degradation products of lipid peroxidation, its content can be measured by the reaction with thiobarbituric acid. An increase in free radicals could cause overproduction of MDA (Gawełet al., 2004). Therefore, MDA can be assessed as an oxidative stress marker to detect the degree of oxidative stress of organisms (Amin et al., 2018; Tsikas, 2017). Previous studies have shown that the changes of antioxidant enzyme activities and MDA content are closely related to the reactions of oxidative stress. Some pollutants, such as microplastics, imidazolyl, and copper, could induce significant changes of antioxidant enzyme activities and MDA content in planarians (Gao et al., 2022; Wang et al., 2020; Zhang et al., 2016a). In this study, the activities of SOD, CAT, and MDA content increased after SDS treatment, indicating that SDS might cause the production of ROS, which in turn activated the corresponding antioxidant enzymes and led to an increase in their activities. Due to the inability of ROS to be completely cleared, it caused oxidative damage to cells, leading to lipid peroxidation in planarians.

In addition to the changes of antioxidant enzyme activities, many compounds can cause damages to the DNA of organisms. These damages include DNA strand breaks, base modifications, and substitutions (Pandey, Nagpure & Trivedi, 2018). RAPD is a sensitive method for detecting DNA damage at the molecular level, and the degree of DNA damages can be reflected by the GTS values (Aksakal & Esim, 2015). In this study, SDS led to the changes of GTS values of planarians, which might be related to DNA damages caused by the changes of oligonucleotide sites, and the breakage, insertion, or deletion of DNA fragment (Tofalo & Corsetti, 2017; Zhang et al., 2017; Zhang et al., 2016b). The GTS values of the treatment groups decreased with the increasing of SDS concentrations and the extension of exposure time, indicating that SDS had genotoxicity to planarians in a time and dose-related manner. Some similar studies regarding genotoxicity of 8-hydroxyquinoline to Misgurnus anguillicauatus, furacilin to Euplotes vannus, and 1,3-methylimidazole to D. japonica also showed that the genotoxicity caused by drugs to organisms also had certain correlation with time and dose (Nan et al., 2013; Zhang et al., 2016b; Zhou et al., 2011).

Apoptosis is a strictly controlled cell suicide characterized by nuclear condensation, cell shrinkage, membrane blebbing, and DNA fragmentation (Bertheloot, Latz & Franklin, 2021; Majtnerová & Roušar, 2018). Caspase3 is a member of the cysteine family and plays a vital role in the progress of apoptosis (Gong et al., 2022; Lei et al., 2022). Studies have shown that aspirin could reduce the level of caspase3 protein and inhibit cell apoptosis, microplastic could induce the expression of caspase3 gene and promote cell apoptosis in planarians (Gao et al., 2022; Liang et al., 2022). In the present study, the expression level of Dj-caspase3 was upregulated in the treatment groups, and gradually increased with the extension of treatment time, suggesting that SDS could induce apoptosis in planarians. PIWI proteins have broader functions in many vital biological processes including cell proliferation, differentiation, and survival. PiwiA and piwiB are members of the PIWI protein family (Kashima, Agata & Shibata, 2020; Ponnusamy et al., 2017). It has been reported that piwiA and piwiB are specifically expressed in neoblasts and encode PIWI proteins in cytoplasm and nucleus respectively (Kashima, Agata & Shibata, 2020). The main function of PiwiA is to maintain cell proliferation, and piwiB is involved in the regulation of neoblasts differentiation (Reddien et al., 2005; Shibata et al., 2016; Tharp & Bortvin, 2016). PCNA is a key factor in the process of DNA replication, and the expression of PCNA in all organisms is related to cell proliferation and DNA synthesis during genome replication in S phase of cell cycle (Orii, Sakurai & Watanabe, 2005; Strzalka & Ziemienowicz, 2011). Therefore, Dj-piwiA, Dj-piwiB and Dj-PCNA can be used as marker genes to detect the effects of pollutants on the proliferation or differentiation of planarian cells. Here, our results showed that the expression levels of Dj-piwiA, Dj-piwiB and Dj-PCNA were downregulated in the treatment groups, especially on the third and the fifth day. We speculated that with the extension of exposure time, SDS might decrease the proportion of mitotic stem cells and consequently restrain the proliferation or differentiation of planarian cells. CyclinB is an important regulator of cell cycle, which is responsible for the transition from G2 phase to M phase in the cell cycle (Van Wolfswinkel, Wagner & Reddien, 2014; Zhong et al., 2019). The inhibition of proliferation in the regenerating planarians treated by aspirin might be related to the abnormal cell cycle caused by the reduced expression of cyclinB (Liang et al., 2022). A similar study showed that downregulation of G2/mitotic-specific cyclinB could constrain proliferation, induce apoptosis, and trigger autophagy in nasopharyngeal carcinoma cells (Xie et al., 2019). In this study, the expression level of Dj-cyclinB significantly decreased in the treatment groups, indicating that SDS might interfere with cell cycle progression via downregulation of cyclinB, and then lead to the inhibition of cell proliferation. DNA integrity is crucial for maintaining the homeostasis of planarian tissues. RAD51 is an essential component in maintaining the genomic stability and repairing DNA double strand break, so its encoding gene can be used as a marker gene for detecting the degree of DNA damages (Barghouth et al., 2019; Bonilla et al., 2020; Prado, 2021). In this experiment, the expression level of Dj-RAD51 was significantly downregulated in the treatment groups, suggesting that SDS could reduce the ability of DNA repair, leading to gene mutation and genomic instability in planarians.

Conclusions

Taken together, SDS has acute toxicity and genotoxicity to planarian D. japonica. SDS of 0.5 mg/L and 1.0 mg/L could induce oxidative stress and genetic toxicity in planarians.1.0 mg/L SDS upregulated the expression of apoptosis-related gene, downregulated the expression of genes related to cell cycle, cell proliferation and DNA repair ability. These results indicate that SDS has toxic effects on freshwater planarians and potential hazards to the aquatic environment. Our study provides a theoretical basis for the risk assessment and management of SDS, as well as for the protection of aquatic organisms.

Supplemental Information

Table S1 Sequences of 13 primers used in RAPD analysis

Click here for additional data file.

Table S2 Primers used in qPCR

Click here for additional data file.

Data S1 Raw data of LC50, enzyme activities and qPCR experiment

Click here for additional data file.

Data S2 RAPD profiles of planarian D. japonica

Note: 0.5–1; 0.5–3; and 0.5–5 represent D. japonica exposed to 0.5 mg/L SDS for 1, 3, and 5 days respectively. 1–1; 1–3; and 1–5 represent D. japonica exposed to 1.0 mg/L SDS for 1, 3, and 5 days respectively.

Click here for additional data file.

Abbreviation Index

SDS Sodium dodecyl sulfate

LC50 Semi-lethal concentration

SOD Superoxide dismutase

CAT Catalase

MDA Malondialdehyde

RAPD Random amplified polymorphic DNA

PCR Polymerase chain reaction

GTS Genomic template stability

ROS Reactive oxygen species

qPCR Fluorescent quantitative PCR

Dugesia japonica D. japonica

Additional Information and Declarations

Competing Interests

Author Contributions

Data Availability

The authors declare there are no competing interests.

Minmin Feng conceived and designed the experiments, performed the experiments, analyzed the data, prepared figures and/or tables, authored or reviewed drafts of the article, and approved the final draft.

Zhenbiao Xu conceived and designed the experiments, authored or reviewed drafts of the article, and approved the final draft.

Dandan Yin analyzed the data, authored or reviewed drafts of the article, and approved the final draft.

Zelong Zhao performed the experiments, prepared figures and/or tables, and approved the final draft.

Xiuyuan Zhou analyzed the data, authored or reviewed drafts of the article, and approved the final draft.

Linxia Song conceived and designed the experiments, authored or reviewed drafts of the article, and approved the final draft.

The following information was supplied regarding data availability:

The raw data is available in the Supplemental Files.

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
