# Peer review of "Toxic effects of sodium dodecyl sulfate on planarian Dugesia japonica"

_PeerJ, doi:10.7717/peerj.15660_

## Round 0.1 · original submission · Major Revisions

Please revise the manuscript according to the reviewers' comments.

Reviewer 1 ·

Basic reporting

This research provides valuable information on the toxic effects of SDS on planarian D. japonica, including the induction of oxidative stress and genotoxicity, as well as alterations in gene expression that affect cell proliferation, differentiation, and DNA repair. These findings contribute to our understanding of the potential risks associated with SDS exposure in aquatic environments.

Experimental design

It is very well-designed as to support the results.

Validity of the findings

Please see the attached PDF.

Additional comments

Please see the attached PDF.

Annotated reviews are not available for download in order to protect the identity of reviewers who chose to remain anonymous.

Reviewer 2 ·

Basic reporting

The introduction provided enough information to know the importance of the study and supply the readers with enough context as to why the research was conducted. However, there are inconsistencies in the narration (e.g. First person view). I highly recommend the authors to submit their manuscript for language editing.

The introduction could be improved. The flow and structure of the paper can be revised to make it more clear.

The references used in the introduction could be improved.

The figures are clear with appropriate labels.

The results provided match the hypotheses of the research.

Specific line comments:

Abstract:

11-12: Maybe enhance this portion of the abstract by adding 1-3 sentences about the methodology being used (e.g. what tests were done) before discussing the results of the acute toxicity test to give an insight as to why these tests were used and what it is used for.

14-17: This portion talked about a different type of analysis. This can be clarified after discussing the methods beforehand to give the readers a bit more background as to why such tests were used.

21-22: To clarify, dose and time-related manner, is this referring to the dose-response curve?

22-27: It seems to me that the methods were used to evaluate the toxicity of SDS up to the genetic level. Maybe mention this somewhere in the methods portion (see line 11-12 comment).

Introduction:

29: Add s to surfactants. (Surfactants are a kind of…). Maybe use amphipathic, instead of hydrophobic and hydrophilic.

32-33: To clarify, surfactants are degradable, is this through treatment? Or is it biodegradable? Does this refer to surfactants in general? Or only anionic surfactants?

34-37: Please find a reference for this line. (Sodium dodecyl sulfate is a common anionic surfactant…)

37:-38: Maybe improve the sentence by changing this phrase (SDS is closely related to human life…). Or if not, please provide a bit more detail as to why it's related to human life.

37-43: Perhaps you can restructure this paragraph to make it shorter by writing the applications of SDS on various fields and its content would vary, how and why it varies.

44-46: Please add 2-3 references more for this sentence (due to discharge of cleaning products…) to justify its environmental risk to aquatic life.

45-46: In Bondi’s paper, the authors stated that formulations of SLS from consumer products is “diluted and are not necessarily moderately toxic, and in fact, can be non-toxic to aquatic life” (p. 30). I think this reference should be changed or more references should be added especially in line 44, the source of SDS written in the paper is from domestic wastewater from the discharge of cleaning products. I think it would be better if the authors could find a different reference to support this claim, especially that millions of people in the world use these products and it maintains its concentrations in the environment from urban effluents, hence highlighting the importance of SDS pollution in aquatic environments.

46-58: Maybe restructure and shorten this portion to make it more organized: I suggest to arrange it according to:
-SDS has been linked to detrimental toxic effects to aquatic organisms (cite references 2-3 more).
-Its mechanism as to why its toxic (how does it affect aquatic life).

59-68: This paragraph is out of place. Maybe transfer or restructure it to make it more cohesive with the previous paragraph. You can shorten this paragraph by highlighting the importance of RAPD and its applications on genotoxicity studies, cite references where it has been used (no need to discuss the results of the studies), and the gaps.

69-83: Transfer this paragraph before the RAPD paragraph.

69-71: Why are planarians important members of freshwater ecosystems? What is their role in these ecosystems? Discuss in 1-2 sentences with citations.

71-75: First sentence and second sentence are disconnected. Perhaps revise this to make it clearer.

75-77: Maybe transfer this after line 71 to justify the importance of planarians outside ecosystem functioning and more of its importance in the determination of toxicity studies.

77-81: There is no mention of RAPD. Maybe include this, since qPCR was mentioned to determine DNA repair ability. To be consistent, maybe only mention the main aim of this study without mentioning how it is being done since it can be discussed in the methods portion.

82-83: Add s to water ecosystem(s). To clarify, is this freshwater or just water? You may say water ecosystem(s) or freshwater ecosystem.

Tables:

For Tables 1 and 2, I think it would be better if these tables were moved under appendices and cited as an Appendix (e.g. Appendix 1).

Experimental design

The article is an original primary data research.

The research question was appropriately mentioned. The statement of the research gaps and the impact of the research could be improved.

The study involved various methods to measure the impacts of SDS. However, references for the methods used were not included in the write-up.

The methodology was not described in full detail. There were a lot of missing information that might affect future replicability. I highly encourage the authors to provide more information regarding this.

I encourage the authors to add a brief explanation on the principle behind each method to give the readers and future researchers about the background on why such parameter is important, how it is quantified, and its implications for the result.

Specific Line comments:

86: What is the purity of the SDS used?

86-87: Please spell out SOD, CAT, and MDA.

86-92: Before mentioning where these materials were obtained, maybe it’s better to include the kits and for which analysis it would be used to guide the readers/future researchers what the material is and what it is used for.

88: Please clarify if EZNA was used? Or simply to inform the readers where the product was developed?

94: When you say cultured, is it propagated in the lab? Or environmental sampling was done first to obtain the organism and then further developed and propagated in the lab? Where did the planarians originally come from? Please clarify.

Maybe mention how long the organisms were kept in the lab before the experiment started.
To clarify, are the planarians used for the experiment adult planarians?

95: Why is Montjuïc water used for this study? Why not distilled water or deionized distilled water? Please discuss in 1-2 sentences.

100: What is the basis for the SDS concentrations? Please cite your references or justify why this approach was done. How were the levels determined? And why 1 cm? Why was the acute toxicity test done? Please discuss in 1-2 sentences with references.

103-104: Please revise the sentence (changed the treated solution…)

104: Why was the experiment repeated 3 times? Please explain this portion.

104-105: Please revise the sentence (regression equation was obtained…). Why was the regression equation computed? What is its purpose? Please explain in 1-2 sentences.

105: What is “odds unit”? and why was it used?

106: To clarify, the LC50 was calculated using the regression equation and the odds unit?

109: Why is it important to determine the antioxidant enzyme activity? Please discuss in 2-3 sentences with references. What was the basis of the SDS solution and the number of days? Please cite references or justify why this method was used.

111- Please discuss in more detail how sampling collection was done after SDS exposure. When did the experiment end, what was done to process the samples? Please cite the method used for this section.

113- Is it possible to include the instructions from the kit? Perhaps add 1-2 sentences about it or include this in the supplementary info.

116-119: Explain genomic DNAs in 1-2 sentences and why it is important to study this parameter? What did this study use 0.5 and 1.0 mg/L concentrations? What is the reference for this method? What do you mean by OD260/OD280? Please spell out and discuss its importance.

121: Please spell out RAPD. Discuss why RAPD is essential to be determined in this study in 1-2 sentences.

122: Please spell out PCR. Please cite the reference used for this method.

131: Please spell out GTS. What is the importance of the GTS for this study? Discuss in 1-2 sentence?

136-142: Add a transition sentence in the first part of the paragraph. (After x was done, RNA extraction and qPCR was done…). Add why RNA extraction and qPCR was conducted and its purpose in 1-2 sentences. Why was the collection done only for the 1.0 mg/L SDS group? Please explain. Please cite your references for this method.

144: How was the comparison of the results done? Was it through assigning letters to indicate significant differences? How was it done? Please add it accordingly.

Validity of the findings

This study can provide a lot of information as to how aquatic organisms function from exposure to SDS.

All underlying data for this study was provided.

The conclusions of this paper can be improved.

It would be better if the authors included implications in the discussion (X upregulated Y leading to Z).

Specific Line Comments:

Results:

150- Add transition sentence. (e.g. Toxicity of SDS was done through…).
R2 was not mentioned in the methods. Please add this accordingly under statistical analysis.

157- Please spell out acronyms in every new paragraph. (SOD, CAT, and MDA).

163- Clarification: …increased firstly, and then decreased from the first day? Is it not during the second day? Or do you mean CAT activity initially spiked upon exposure, and then gradually decreased? Please clarify.

156-168: Maybe it would be better to also include statistical differences in the discussion while providing the trend of the figures to provide the readers an explanation that such trend is significant or not significant.

175- Please spell out acronyms.

190- The parameters discussed here were not discussed in the methods section as well as how and why they were collected. Please adjust this portion accordingly.

200-201: These parameters should be included in the methods section. Please revise accordingly.


Discussion:

205- Li determined? Is this a reference? If so, please add the date and use a proper citation format. E.g. Li (2008) determined…

208-210: Maybe this sentence (In order to accurately…) should be added as the first sentence under this paragraph.

211- holbrooki. Perhaps it would be better to mention what the study is about (e.g. The study of Nunes et al. (2005) determined the LC of various organisms including shrimp [insert scientific name])

212-214: Why is it higher? Please expound.

220- Please spell out MDA.

221- Please add a discussion about the levels of MDA quantified in the study to justify its importance as a biomarker.

215-228: Perhaps a better structure for this discussion would be
-Importance of the parameter, the general trend (e.g. higher, lower levels), an explanation of the trend with references, and its potential use for future studies.

230- Please spell out RAPD, GTS.

237-241: Maybe revise the sentence to (A similar study regarding…)

243- Should Caspase3 be written as Caspace3 in italic? Please check for consistency.

245- How is diflubenzuron and cryptotanshinone relevant to be mentioned for this study? How is this reference related to the current study? Maybe revise this sentence to make it clearer.

249-250: Maybe it would be better to add a transition sentence before discussing piwiA and piwiB to contextualize this part.

262-264: Maybe restructure this sentence as (A similar study using…).

275-280: Perhaps move this under conclusions.

Conclusions:

284-286: Please add a few implications from the results of this study.

I think it would be better to include in the conclusions, the level of SDS that causes the various impacts to the organism (e.g. 0.5 mg/L is enough to elicit a response from Planaria that lead to…)

Additional comments

I highly commend the authors for conducting a very comprehensive assessment of SDS and its impact on planaria. This study provides the information that SDS toxicity is occurring, and it provides proof that SDS is toxic to aquatic life forms and that there is a need for future studies regarding SDS toxicity. I also applaud the authors for measuring the impacts of SDS up to the gene level, which I think is lacking in SDS studies. A job well done!

Annotated reviews are not available for download in order to protect the identity of reviewers who chose to remain anonymous.

·

Basic reporting

This study explored the SDS toxic effect on planarians through LC50 determination, SOD, CAT, MDA, and RAPD analysis. The results of this study provide data reference for the toxicity of SDS to aquatic organisms.
The article is well organized, the introduction and background are sufficient, figures and tables are of good quality. Raw data is provided.
Modifications are suggested as follows:
1) The general knowledge of ROS, CAT, and MDA presented in the discussion section (lines 219-231) can be moved to the introduction.
2) The English language of the entire manuscript should be improved. Examples:
a. Lines 105-106: “Changed the treated solution and calculated the mortality of planarians every day.”
b. Lines 112-113: “Those cultured in Montjuïc water were as control.”
c. Lines 124-128: “Each PCR reaction was conducted in a mixture of 25 μL containing 20 ng genomic DNA, 0.2 μmol/L primer, and 12.5 μL 2×Taq PCR StarMix. Amplifications were implemented in a DNA thermocycler (TC-XP, Hangzhou Bioer Technology Co., Ltd, China) with PCR programmed for 5 min at 94℃, 40 consecutive cycles, each consisting of 1 min at 94℃, 1 min at 37℃, and 2 min at 72℃, then followed by 10 min at 72℃ for the final extension.”
d. Lines 141-143: “The qPCR was implemented in Light Cycler 480 System (Roche Diagnostics, Switzerland) with PCR programmed for 30 s at 95℃, 40 consecutive cycles, each consisting of 5 s at 95℃, 10 s at 58℃, and 15 s at 72℃.”
e. Line 178-179: “Clear and distinguishable bands were obtained after the RAPD-PCR products were subjected to agarose gel electrophoresis.”

3) At lines 21-22, “RAPD analysis showed that the genomic template stability (GTS) values of planarians were 67.86%, 64.29%, 58.93%, and 64.29%, 60.71%, 48.21%, respectively.” It’s not clear what are these values for, please clarify the analysis conditions.
4) Please clarify the content of SDS at lines 41-42 is based on volume or weight.

Experimental design

1) It’s not clear how the LC50 was determined at different exposure days, suggest adding more details on the calculation of LC50.
2) One-way ANOVA analysis is not sufficient for the SDS dose and exposure time analysis, suggest doing two-way or two-way with interaction analysis. See detail comments in the results section.

Validity of the findings

1) I suggest adding the notes for terms of x and y in Table 3.
2) The data can not support the results in the section “Effects of SDS on oxidative stress”. Only One-Way ANOVA analysis is done for exposure dose (control to with SDS). There’s no evidence to show that there’s a significant increase in SOD, CAT, and MDA content with the exposure time. Therefore, no data support the statements such as: “With the extension of SDS exposure time, SOD activity in 0.5 mg/L SDS treatment group increased significantly, reached the highest on the fifth day. In the 1.0 mg/L SDS treatment group, SOD activity increased on the first day, continued to increase on the third day, and returned to the same level as the control group on the fifth day (Figure 1A).”
3) Discussion section:
Li’s (Li 2008) results showed the LC50 of SDS to planarian was much lower than the result of 3.76 mg/L in this study. Please clarify why there’s no big difference and how the current study can provide more accurate results compared to previous studies in reference.

---

## Round 0.2 · Minor Revisions

Some minor revisions are still needed.

Reviewer 1 ·

Basic reporting

This research provides valuable information on the toxic effects of SDS on planarian D. japonica, including the inductcion of oxidative stress and genotoxicity, as well as alterations in gene expression that affect cell proliferation, differentiation, and DNA repair. These findings contribute to our understanding of the potential risks associated with SDS exposure in aquatic environments.

Experimental design

It is well-designed and well-supported using solid data and reference.

Validity of the findings

The findings are very interesting and informative supported by strong data.

Additional comments

Great job done by all your dedicated team! The revision pretty much solved all my questions, it is in perfect status to move to the final publication. Cheers! Congrats!

Reviewer 2 ·

Basic reporting

This version of the manuscript is much more organized and clear. Each sentence is well-justified with relevant literatures. I highly commend the authors for taking the time to address every single comments.

I only suggest that the authors submit their manuscript for language editing, preferably by a native anglophone. I observed inconsistencies (e.g. commas) which will be corrected after language editing.

Experimental design

This version of the manuscript is more detailed in terms of the materials and methods. Each methodological step has been justified as to why such approach was used. Details on each variable have also been added to provide context as to why each parameter is important.

Validity of the findings

No further comments

Additional comments

Abstract:
- No further comments

Introduction:
36-39: Please move although most surfactants paragraph to line 54 and combine with line 54-55

74-82: I think it would be more appropriate to transfer this paragraph under materials and methods to justify the use of planarians before the discussing the method.

131: Insert name of the kits (____)

137: Im assuming its the same kit from line 131? If not, please add the name for each one to prevent confusion.

154-156: Insert as an equation format in the paragraph 

GTS (%)… eq. 1

Where:
A- number of polymorphic bands


Results:

-no further comments.

Discussion:

242- Please remove (Li 2008) after PFOA as it has been cited already in the beginning of the sentence

242-248: I think it would be better to put a transition sentence before discussing the results of other studies to better contextualize the results of your study.

267-271: I suggest moving this paragraph in the first part of the entire passage (Line 249).

306: I think it would be better to discuss why Dj-piwiA, Dj-piwiB and Dj-PCNA were down regulated specifically on the 3rd and 5th days. What happens during these particular time frames?

·

Basic reporting

This study explored the SDS toxic effect on planarians through LC50 determination, SOD, CAT, MDA, and RAPD analysis. The results of this study provide data reference for the toxicity of SDS to aquatic organisms. The article is well organized, the introduction and background are sufficient, figures and tables are of good quality. Raw data is provided.

All comments have been well addressed. No further comments.

Experimental design

All comments have been well addressed. No further comments.

Validity of the findings

All comments have been addressed.

One more comment:
By convention, a result of p-value is considered statistically significant if p < 0.05, highly significant if p < 0.01, and very highly significant if p < 0.001. But it is unusual to use terms such as 'extremely significant' in scientific articles results and discussions. Suggest revising the terms which are too subjective in the results and discussion sections.

Additional comments

None

---

## Round 0.3 · accepted · Accept

Congratulations! Your paper can be accepted now.